# The Peripheral Immune Traits Changed in Patients with Multiple System Atrophy

**DOI:** 10.3390/brainsci13020205

**Published:** 2023-01-26

**Authors:** Zhenxiang Gong, Rong Gao, Li Ba, Yang Liu, Hongyan Hou, Min Zhang

**Affiliations:** 1Department of Neurology, Tongji Hospital, Tongji Medical College, Huazhong University of Science and Technology, Wuhan 430030, China; 2Department of Laboratory Medicine, Tongji Hospital, Tongji Medical College, Huazhong University of Science and Technology, Wuhan 430030, China

**Keywords:** multiple system atrophy, immunity, dysregulation, central nervous system, peripheral blood

## Abstract

A growing body of evidence suggests immune involvement in the pathology of multiple system atrophy (MSA). Research on detailed peripheral immune indices, however, is relatively sparse, and is one of the intriguing aspects of MSA yet to be elucidated. A total of 26 MSA patients and 56 age-and sex-matched healthy controls (HC) were enrolled in the current case-control study to delineate the peripheral immune traits of MSA patients. The ratio of CD4^+^/CD8^+^ T cells, natural killer cells, CD28 expression on both CD4^+^ T cells and CD8^+^ T cells increased in MSA patients compared to HC, but CD8^+^ T cells and active marker (HLA-DR) expression on total T cells decreased (*p* < 0.05). This study sheds light on the dysregulation of cellular immunity in MSA, pointing to future mechanistic research.

## 1. Introduction

Multiple system atrophy (MSA) is an adult-onset, progressive neurodegenerative disease characterized by varying degrees of parkinsonism, cerebellar ataxia, and dysautonomia [1]. MSA is classified clinically into two types based on the most prominent symptoms: parkinsonian MSA (MSA-P) and cerebellar MSA (MSA-C) [1]. The argyrophilic filamentous glial cytoplasmic inclusions (GCIs), whose core component is misfolded alpha-synuclein (α-synuclein), are indicative of MSA [2]. However, the pathologic mechanisms by which MSA develops and progresses are poorly understood. No disease-modifying treatments are currently available to delay disease progression in MSA.

Potential pathogenic mechanisms previously reported include cell-to-cell spreading of α-synuclein, oxidative stress, mitochondrial dysfunction, and neuroinflammation [3,4]. Among all possible causes of MSA pathology, the neuroinflammation theory has received attention in the last decade. MSA patients had higher levels of proinflammatory cytokines in their frontal cortex and cerebrospinal fluid (CSF) [5,6,7,8]. Both astrocytes and microglia are involved in the abnormal deposition of α-synuclein, and GCI may trigger a cascade of inflammatory immune responses that result in neuronal cell degeneration and necrosis [9].

In neurodegenerative diseases, there has recently been an increase in evidence of immune crosstalk between the central nervous immune system and the peripheral immune system [10,11]. The concept of immune privilege and deprivation of peripheral immune cells in the central nervous system (CNS) has been disproven [12]. The presence of CD4^+^ and CD8^+^ T cells in the substantia nigra of a Parkinson’s disease mouse model and of patients, altered cytokine profile in serum, as well as the proximity of infiltrating T cells to major histocompatibility complex-expressing microglia, suggests the circulation and migration of peripheral immune cells to sites of inflammation [13,14]. Antigen-specific immune cell migration from the periphery to the CNS, and consequent immune cell interactions with resident glial cells, affect neuroinflammation and neuronal survival [15]. Thus, studying peripheral immune abnormalities helps us to understand immune-related mechanisms in neurodegenerative diseases. According to preliminary studies, peripheral immune abnormalities are involved in the pathology of MSA [16,17,18,19]. T cell infiltration in the putamen and substantia nigra was found to be increased in MSA patients [19]. According to Cao et al., the ratio of CD4^+^/CD8^+^ T cells was higher in male MSA patients compared to healthy controls (HC) [16]. In the current study, the ration of CD4^+^/CD8^+^ T cells was also higher in MSA patients compared to HC. In another study, Zhang et al. reported that a high neutrophil-to-lymphocyte ratio (NLR) predicts short survival in MSA patients [17]. Madetko et al. found that the NLR was positively correlated with disease duration in MSA-P patients [18]. However, previous researchers either only explored the changes of blood routine test, or simply divided the peripheral T cells into CD4^+^ T cells (helper T cells: Th) and CD8^+^ T cells (cytotoxic T cells: Tc). Other T cell subsets, such as CD28^+^ T cells and regulatory T cells (Tregs), were recently shown to be closely associated with α-synuclein pathology [20,21]. A detailed examination of these T cell subsets in MSA patients is currently lacking. A thorough examination of changes in peripheral immune traits in MSA patients provides clues for future mechanistic studies. Therefore, this study aims to: (1) provide a comprehensive analysis of 34 kinds of peripheral immunity-related indices in patients with MSA; and (2) review previous studies and propose new research directions.

## 2. Materials and Methods

### 2.1. Study Design, Patient Recruitment, and Ethics Approval

The study flow chart is displayed in Figure 1. In this case-control study, 26 MSA patients and 56 age- and sex-matched HC were consecutively recruited at the Department of Neurology, Tongji Hospital, Tongji Medical College, Huazhong University of Science and Technology, from May 2019 to May 2022. According to the 2008 diagnostic criteria, all patients with MSA had disease onset after the age of 30 years and were diagnosed at probable or definite levels [1]. Participants were excluded if they had a family history of neurodegenerative disorders, immunodeficiency or autoimmune diseases, or if they were taking immunomodulatory medications. Serum biochemical tests (normal level of C-reactive protein and erythrocyte sedimentation rate) revealed no evidence of systemic inflammation in any of the participants. Demographic information of all subjects and clinical information of MSA patients were recorded. This study was approved by the ethics committee of Tongji Hospital, Tongji Medical College, and the Huazhong University of Science and Technology (TJ-IRB20220829). All participants provided written informed consent, and the study was carried out in accordance with the Helsinki Declaration [22].

### 2.2. Blood Sample Collection and Flow Cytometry

Heparinized venous blood samples were collected by venipuncture from MSA patients and HC between 6:00–7:00 a.m. The absolute numbers and phenotypes of lymphocyte subsets were then determined using flow cytometry. The percentages and absolute numbers of CD4^+^ T cells, CD8^+^ T cells, B cells, and natural killer (NK) cells were determined using TruCOUNT tubes and BD Multitest 6-color TBNK Reagent Kit (BD Biosciences Pharmingen, San Diego, CA, USA) following the manufacturer’s instructions. In brief, 50 µL of whole blood was incubated for 15 min at room temperature with a six-color TBNK antibody cocktail. Samples were analyzed with a FACSCanto flow cytometer and FACSCanto clinical software after adding 450 µL of FACS lysing solution (BD Biosciences Pharmingen). The following mAbs were added to 100 µL whole blood. The antibodies in panel 1 were anti-CD45-PerCP (BD Biosciences Pharmingen, 2D1), anti-CD3-APC-H7 (BD Biosciences Pharmingen, SK7), anti-CD4-V450 (BD Biosciences Pharmingen, RPA-T4), anti-CD8-PE/Cy7 (BD Biosciences Pharmingen, SK1), anti-CD28-PE (BD Biosciences Pharmingen, L293), and anti-HLA-DR-APC (BD Biosciences Pharmingen, L243). The antibodies in panel 2 were anti-CD45-PerCP (BD Biosciences Pharmingen, 2D1), anti-CD3-APC-H7 (BD Biosciences Pharmingen, SK7), anti-CD4-BV510 (BD Biosciences Pharmingen, SK3), anti-CD45RA-FITC (BD Biosciences Pharmingen, L48), anti-CD8-PE/Cy7 (BD Biosciences Pharmingen, SK1), anti-CCR7-PE (BD Biosciences Pharmingen, 3D12), anti-CD25-APC (BD Biosciences Pharmingen, 2A3), and anti-CD127-BV421 (BD Biosciences Pharmingen, HIL-7R-M21). The antibodies in panel 3 were anti-CD38-FITC (BD Biosciences Pharmingen, HB7), anti-CD19-PE/Cy7 (BD Biosciences Pharmingen, SJ25C1), anti-CD27-PerCP (BD Biosciences Pharmingen, 2D1) and CD45-V500C (BD Biosciences Pharmingen, 2D1), and anti-IgD-APC (BD Biosciences Pharmingen, IA6–2). The total percentage of T cells (%), B cells (%), and NK cells (%) were calculated as the proportion of CD3^+^CD19^–^ cells, CD3^–^CD19^+^ cells, and CD3^–^CD16^+^CD56^+^ cells in lymphocytes, respectively. The total percentage of helper T cells (Th%), cytotoxic T cells (Tc%), activated T cells (%), Treg (%), natural Treg (%), and induced Treg (%) were calculated as the proportion of CD3^+^CD4^+^ T cells, CD3^+^CD8^+^ T cells, CD3^+^HLA-DR^+^ T cells, CD3^+^CD4^+^CD25^+^CD127low^+^ T cells, CD45RA^+^CD3^+^CD4^+^CD25^+^CD127low^+^ T cells, and CD45RO^+^CD3^+^CD4^+^CD25^+^CD127low^+^ T cells in total T cells, respectively.

### 2.3. Serum Cytokine Detection

A chemiluminescent immunometric assay was used to detect serum interleukin-2 receptor (IL-2R), IL-1β, IL-10, IL-8, and tumor necrosis factor-α (TNF-α) (IMMULITE 1000 Analyzer, Siemens). Serum IL-6 was measured using the electrochemiluminescence method (Roche Diagnostics, South San Francisco, CA, USA) [23].

### 2.4. Search Strategy for Literature Review

In order to learn more about peripheral immunity in MSA, we summarized data from previously conducted clinical investigations. We searched the PubMed database for studies that had been published prior to November 2022 describing the changes in the peripheral immune system in MSA patients. The search terms used were “(multiple system atrophy [Title/Abstract]) AND (lymphocyte OR cytokines OR NLR OR leukocyte OR inflammasome)”. Studies were considered for inclusion if they met both of the following criteria: (1) the study had to be written in English with its entire text publicly accessible; (2) the study had to be focused on the changes of peripheral immunity in MSA patients. Reviews, clinical trials, and studies focused on other topics were excluded. In total, 15 studies were summarized for this review (Table 1).

### 2.5. Statistical Analysis

Continuous variables were either represented as mean (standard deviation, SD) or median (25% quartile and 75% quartile). Depending on the distribution of data, differences between the two groups were analyzed using Student’s *t*-test or Mann–Whitney U test. Categorical variables were represented as frequencies and were compared using the chi-square test. In the partial least squares discriminate analysis (PLS-DA), samples with different peripheral immune characteristics have different aggregation regions in the two-dimensional coordinate system, which was further validated using a permutation test (200 permutations). The variable importance in the projection (VIP) values were calculated to screen for major immune indices contributing to the peripheral immunity differences between MSA and HC. *p* < 0.05 indicated statistical significance. All statistical analyses were performed with SPSS (IBM, version 26.0) and SIMCA-P (Umetrics, version 14.1) software. Graphs were generated using GraphPad Prism 7.0.

## 3. Results

### 3.1. Demographic and Clinical Characteristics

In this study, age- and gender-matched 26 MSA and 56 HC were enrolled (Table 2). The median age of onset and disease duration for MSA patients was 54 years and 15 months, respectively. Twenty-four patients were diagnosed as MSA-C and two patients as MSA-P, based on their clinical symptoms.

### 3.2. Changes of Peripheral Immune Traits in Multiple System Atrophy Patients

There was no statistically significant difference between MSA and HC in the counts of white blood cells, neutrophils, lymphocytes, monocytes, eosinophils, and basophils. The neutrophil-to-lymphocyte ratio was also comparable between MSA and HC (Table 3). The ration of Th/total T cells, the ratio of CD4^+^/CD8^+^ T cells, natural killer cells, CD28 expression on both CD4^+^ T cells and CD8^+^ T cells were higher in the MSA group (*p* < 0.05), but Tc and active marker (HLA-DR) expression on total T cells was lower (*p* < 0.05) (Table 4). There was no statistical difference in serum cytokine levels between MSA and HC (Table 5). The correlations between clinical characteristics and lymphocyte subsets in MSA patients were estimated. We observed positive correlations between the NK cells (%) and onset age (rs = 0.463, *p* = 0.017), the NK cell counts and onset age (rs = 0.427, *p* = 0.030), and the ratio of CD3^+^CD8^+^HLA-DR^+^ Ts/Ts with onset age (rs = 0.462, *p* = 0.017). No correlation was observed between these indices and disease duration (Appendix A). Additionally, we compared the lymphocyte subsets between males and females in patients with MSA. There was no statistical difference between male and female patients (Appendix A).

We also compared the overall similarity of peripheral immune characteristics between MSA and HC using PLS-DA, which showed discrete distributions, indicating distinctive peripheral immune characteristics (Figure 2A). The permutation test demonstrated that the PLS-DA models were valid (Intercepts: *R*^2^ = 0.109 < 0.3, Q^2^ = −0.182 < 0.05). The top ten peripheral immune indices were screened based on the VIP score (Figure 2B).

## 4. Discussion

We observed an increase in peripheral Th/Tc, natural killer cells, CD28 expression on both CD4^+^ T cells and CD8^+^ T cells, a decrease in CD8^+^ T cells and active marker (HLA-DR) expression on total T cells in MSA patients. The current study provides, to the best of our knowledge, the most extensive assessments of peripheral immunity in MSA patients, offering some fresh evidence for peripheral immune dysregulation in these patients.

### 4.1. Increased Natural Killer Cells and Alpha-Synuclein Pathology

NK cells are one of the important components of human lymphocyte lineages, accounting for about 10–15% of circulating lymphocytes [32]. Besides being the first defense of innate immunity, NK cells are also involved in adaptive immunity by producing a wide variety of cytokines and chemokines, including IFN-γ, TNF-α, GM-CSF, IL-10, IL-5, IL-13, MIP-1α, MIP-1β, IL-8, and CCL5 [33,34]. Evidence from neurodegenerative diseases over the past decade suggests that NK cells play a major role in response to central nervous system inflammation [35,36,37]. Increased peripheral NK cells have been reported in previous studies on Parkinson’s disease [37,38,39,40,41,42,43], which seems to be a consistent finding [44]. Although both MSA and PD are neurodegenerative diseases characterized by aberrant α-synuclein deposition, only one study has looked into the changes in peripheral NK cells of MSA patients [26]. In contrast to earlier reports that showed a similar number of peripheral NK cells between MSA and HC [26], we found an increase in peripheral NK cells in the MSA patients. Systemic depletion of NK cells exacerbated α-synuclein pathology in a mouse model of α-synuclein. Peripheral NK cells efficiently infiltrate into the CNS and are involved in the degradation of α-synuclein aggregation via the endosomal/lysosomal pathway [45]. Our findings regarding the increase in NK cells suggest a potential defense mechanism to clear excess α-synuclein in MSA patients. Positive NK cell staining in the substantia nigra has been reported in patients with PD [45], and postmortem brain tissue in MSA requires a similar pathological evaluation. Another issue that cannot be overlooked is that NK cells in humans displayed dynamic expression of surface markers involved in differentiation, trafficking, and cytotoxicity, and distinct subpopulations of NK cells have varied immune functions [46]. Tian et al. reported an increase in the frequency of high cytotoxic NK subsets (CD56^+^CD16^+^CD57^+^CD28^–^ NK) in PD [47]. Reports have shown that CD56^dim^ NK cells, rather than CD56^bright^ NK subtypes, played a vital role in α-synuclein pathology [48]. There is a lack of investigation on precise NK subsets in MSA patients, and further research is needed to answer these unknown but important questions.

### 4.2. Activated Helper T Cells and Multiple System Atrophy

Our study revealed a higher CD28 expression on both CD4^+^ T cells and CD8^+^ T cells, and a lower active marker (HLA-DR) expression on total T cells in MSA patients, suggesting a disrupted T cell immunity. T cells constitute 65–75% of the total lymphocytes playing vital roles in adaptive immunity [49]. Previous reports have shown infiltrated CD3^+^ T cells in the prefrontal cortex [6], putamen and substantia nigra [19] of postmortem MSA tissue. The immunological reactions between these infiltrated T cells and α-synuclein are intricate [50]. Based on the differential expression of cell surface molecules, T cells can be classified into two categories: Th and Tc [51]. Earlier reports have shown an increased proportion of peripheral CD4^+^ T cells in MSA patients [16]. The infiltrated CD4^+^ T cells were higher than CD8^+^ T cells in postmortem brain tissue of MSA patients [19]. In oligodendroglial α-synuclein viral vector models of MSA, the entry of CD4^+^ T cells into intracranial tissue was significantly more robust than CD8^+^ T cells, indicating the importance of CD4^+^ T cells in MSA pathology [19]. Our study showed a higher ratio of CD3^+^CD4^+^CD28^+^ Th/Th in MSA patients. CD28 present on the surface of T cells is responsible for co-stimulation, and CD3^+^CD4^+^CD28^+^ Th is a subpopulation of activated Th cells [52]. In T cells, CD28 expression induces several intracellular events, including cytokines production and signal transduction, leading to their survival or differentiation [53]. The role of CD28^+^ T cells in MSA is unknown at present. Previous studies have classified PD and healthy controls based on CD28 signaling in Th cells using bioinformatic approaches [21]. On the contrary, cytotoxic T-lymphocyte-associated antigen 4 (CTLA4) negatively regulates T cell responses by competing with CD28 for CD80/86 site [54]. The imbalance of CD28 and CTLA4 leads to high inflammatory tendency and dopamine neuron injury [55], which may contribute to the onset and exacerbation of movement disorders in MSA patients. Being the first to detect the detailed subsets of T cells in MSA patients, our findings about the increased CD3^+^CD4^+^CD28^+^ Th cells necessitate further investigation into the underlying mechanisms.

### 4.3. A Summary of Altered Immune Traits in Multiple System Atrophy

Previous studies have shown that postmortem tissues from MSA patients contain an infiltration of CD3^+^, CD4^+^, and CD8^+^ T cells in the putamen and substantia nigra [19]. Compared to controls, there was an increase in the proportion of CD4^+^ T cells, but no change in the proportion of CD8^+^ T cells [16]. In the current study, the proportion of CD4^+^ T cells and CD4^+^/CD8^+^ T cells increased in MSA patients, while the CD8^+^ T cells decreased in MSA patients. No studies have analyzed NK cells subsets and Tregs in MSA patients. There has not been a single study that has analyzed NK cell subsets and Tregs in MSA patients. The NLR, an inflammatory marker, can predict the severity and prognosis of neurodegenerative diseases such as PD [56], progressive supranuclear palsy [57] and amyotrophic lateral sclerosis [58]. Three studies have reported higher NLR in MSA patients being associated with a poor outcome [17,18,24]. However, our current study did not find a significant difference in NLR between MSA and controls. Previous studies on inflammatory cytokines in peripheral blood were inconclusive [27,28,30]. Our study could not find any significant difference in the inflammatory cytokines between the two groups; however, there was a slight increase in IL-6 in MSA patients, corroborating previous findings in cerebrospinal fluid [5,7,8,29] and postmortem brain issues [6]. Additionally, a previous study reported an increased number of NOD-like receptor thermal protein domain associated protein 3 inflammasome-related microglia in the putamen of MSA [31].

### 4.4. Limitations

There are several limitations in the current study. First, this is a single-center case-control study, so more researches involving multi-center longitudinal cohort studies are needed to understand the dynamic changes of peripheral immune traits throughout disease progression. In this study, just two of the patients had MSA-P. Although MSA-C predominates in the Chinese population, increasing the sample size and distinguishing disease subtypes will help us further elucidate the changes of peripheral immunity in MSA patients. Second, using standardized tools, such as the united multiple system atrophy rating scale (UMSARS), to comprehensively assess the condition of patients with MSA, and taking disease severity into consideration, will help us to better understand the role of dysregulated peripheral immune traits in the progress of MSA. Third, NK cells are highly heterogenous, and activated NK cells secrete a wide variety of cytokines. Some important cytokines and chemokines, such as IFN-γ, MIP-1α, MIP-1β, and RANTES, have not been investigated in our study, calling for a more complete cytokine release assay in future research. Additionally, as α-synuclein pathology is a common feature of PD and MSA, PD patients should be included in further studies to help us explore immune markers that distinguish MSA from other neurodegenerative disorders.

## 5. Conclusions

This is the first study to compare peripheral immune traits comprehensively between MSA patients and HC. Our findings suggest that NK cells, CD28 expression on T cells, and HLA-DR expression on T cells might be involved in the pathology of MSA.

## Figures and Tables

**Figure 1 brainsci-13-00205-f001:**
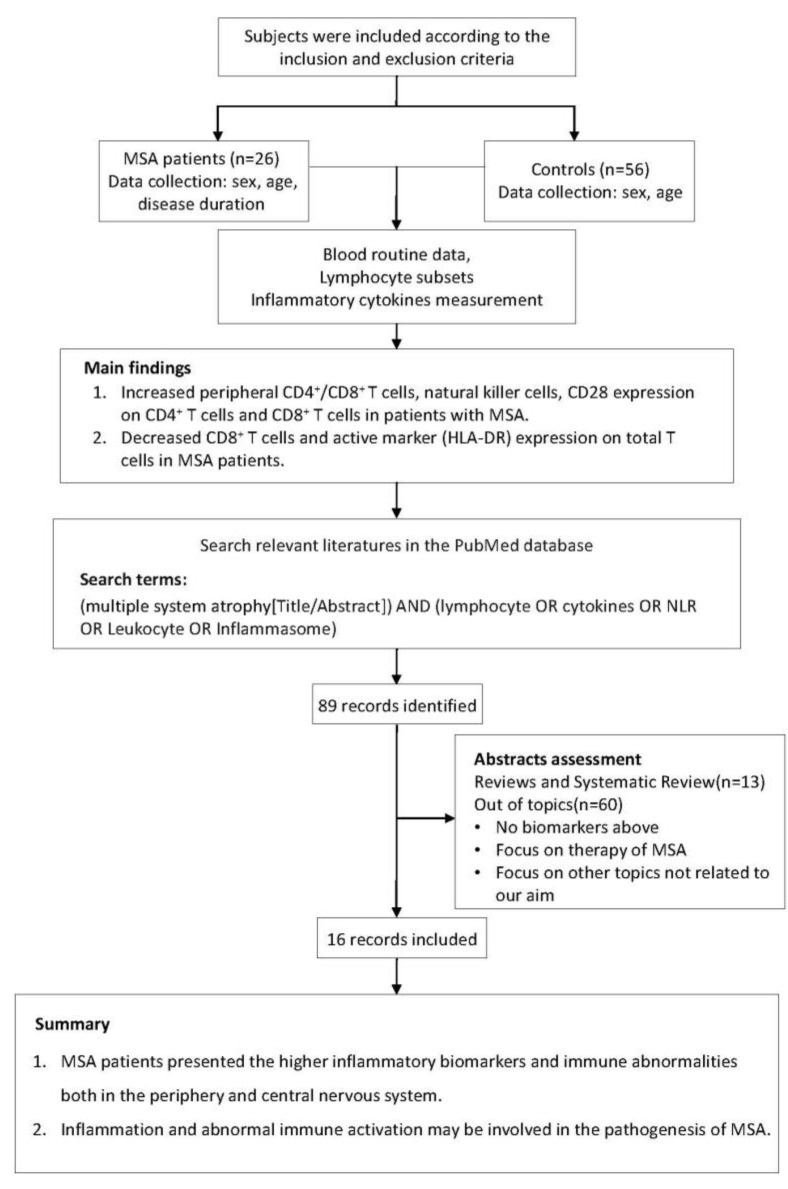
The research flow chart.

**Figure 2 brainsci-13-00205-f002:**
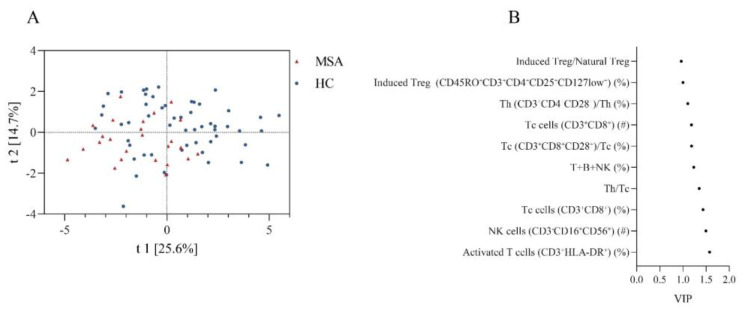
The discrepancy of peripheral immune traits between MSA and HC. (**A**) Obvious separation in the PLS-DA model indicates a different structure of peripheral immune traits between MSA and HC. (**B**) Top 10 peripheral immune indices with highest VIP value between MSA and HC. Abbreviations: MSA: multiple system atrophy; HC: healthy controls; PLS-DA: partial least squares discriminate analysis; VIP: variable importance in the projection.

**Table 1 brainsci-13-00205-t001:** Literature review of studies on immunity traits in patients with MSA.

Reference	Subjects			Immune Traits	Main Findings
	MSA	HC	NDC		
Zhang et al., 2022 [17]	169	163	0	Neutrophil, lymphocyte, monocyte Eosinophilia, basophilia, NLR	Higher NLR in MSA
					Higher NLR was associated with poor survival in MSA
					NLR was not associated with disease progression in 31 MSA patients
Madetko et al., 2022 [18]	28	99	98	Neutrophil, lymphocyte, NLR	Higher NLR in MSA-P
					NLR positively correlated with disease duration in MSA-P
Jiang et al., 2022 [24]	47	124	125	Monocyte/ high-density lipoprotein ratio (MHR) NLR Red cell distribution width/ platelet ratio (RPR)	Higher MHR, NLR, and RPR in MSA
					MHR was positively correlated with the course of 27 MSA-C patients
Matsuse et al., 2020 [25]	34	24	20	Monocyte subsets in serum	Intermediate monocytes (IM) decreased in MSA-C
					Lower IM percentage associated with lower UMSARS scores, shorter disease duration, and milder brainstem atrophy
Rydbirk et al., 2020 [26]	24	46	0	Monocyte subsets in serum	CD14 ^+^ CD16^+^ monocytes decreased in MSA
					No change of CD4^+^, CD8^+^ T cells and NK cells
Cao et al., 2020 [16]	321	321	0	Percentage of CD3^+^, CD4^+^ and CD8^+^ T cells CD4^+^ T cells/CD8^+^ T cells	CD3^+^ and CD4^+^ T lymphocyte increased in MSA
					CD4^+^/CD8^+^ increased in MSA
Kaufman et al., 2013 [27]	14	60	0	CRP, IL-6, IL-2R and TNF-α in serum	IL-6, and TNF-α increased in MSA
					Higher TNF-α associated with less severe motor symptoms and earlier disease stage
Csencsits-Smith et al., 2016 [28]	14	15	25	37 kinds of cytokines and chemokines in serum	GM-CSF, CCL7, and IL-17 decreased in MSA
					IL-4, IL-2, IL-15, and IL-9 increased over time in MSA
Yamasaki et al., 2017 [29]	20	0	27	27 cytokines/chemokines and growth factors in CSF	IL-6, IL-7, IL-12, IL-13, IL-1ra and GM-CSF increased in MSA-C
					FGF, VEGF, IL-1β, IL-2, IL-4, IL-5, IL-8, IL-10, IL-15, MIP-1β, and TNF-α decreased in MSA-C
Hall et al., 2018 [8]	24	50	172	Six kinds of inflammatory biomarkers in CSF (CRP, SAA, IL-6, IL-8, YKL-40 and MCP-1)	Higher CRP and SAA in MSA
					CRP and IL-8 correlated with disease severity in MSA
Starhof et al., 2018 [7]	35	31	85	Eight cytokines (IFN-γ, IL-10, IL-18, IL-1β, IL-4, IL-6, TGF-β1, and TNF-α) and CRP in CSF	No significant difference between MSA and HC
Comptai et al., 2019 [5]	39	15	19	38 kinds of cytokines in CSF	FGF-2, eotaxin, fractalkine, IFN-α2, IL-10, MCP-3, IL-12p40, MDC, IL-17, IL-7, MIP-1β and TNF-α increased in MSA
Kim et al., 2019 [30]	27	20	0	IL-1β, IL-2, IL-6, IL-10, TNF-α and hsCRP in serum	No difference between the MSA and HC
Rydbirk et al., 2017 [6]	19	17	31	18 kinds of cytokines in the dorsomedial prefrontal cortex in brains	IL-2 increased in MSA
					IL-13 and G-CSF decreased in MSA
					Increased MHC class II^+^ and CD45^+^ positive cells, decreased infiltrating CD3^+^ cells in MSA
Li et al., 2018 [31]	11	6	5	NLRP3 inflammasome-related proteins in brains	Increased NLRP3 inflammasome-related microglia in the putamen of MSA
Williams et al., 2020 [19]	3	6	0	CD3^+^, CD4^+^, and CD8^+^ T cells in brains	Increased CD3^+^, CD4^+^, and CD8^+^ T cells in the putamen and substantia nigra of MSA

Abbreviations: CCL: chemotactic cytokines ligand; CSF: cerebrospinal fluid; FGF: fibroblast growth factor; G-CSF: granulocyte colony-stimulating factor; GM-CSF: granulocyte-macrophage colony-stimulating factor; HCL healthy control; hsCRP: high-sensitivity C-reactive protein; IFN: interferon; IL: interleukin; MCP: monocyte chemotactic proteins; MDC: macrophage-derived chemokine; MHC: major histocompatibility complex; MIP: macrophage inflammatory proteins; MSA: multiple system atrophy; MSA-C: MSA with predominant cerebellar ataxia; MSA-P: MSA with predominant parkinsonism; NDC: neurological disease controls; NLR: neutrophil-to-lymphocyte ratio; NLRP3: NOD-like receptor thermal protein domain-associated protein 3; SAA: serum amyloid A; TNF: tumor necrosis factor; UMSARS: unified multiple system atrophy rating scale; VEGF: vascular endothelial growth factor.

**Table 2 brainsci-13-00205-t002:** Demographic and clinical characteristics.

Parameters	Participants		Effect Size	*p* Value
	MSA (*N* = 26)	HC (*N* = 56)		
Age (years)	56.62 (8.37)	53.16 (8.85)	1.673	0.098 ^a^
Female/Male	10/16	29/27	1.264	0.261 ^b^
Age at onset (years)	54 (47, 61)	-	-	-
Duration (months)	15 (12, 36)	-	-	-
MSA-C/MSA-P	24/2	-	-	-

Data presented as mean (SD) or median (25% quantile, 75% quantile). *p* value calculated by Student’s *t*-test ^a^ or chi-square test ^b^. Abbreviations: MSA: multiple system atrophy; HC: healthy controls; MSA-C: cerebellar MSA; MSA-P: parkinsonian MSA; SD: standard deviation.

**Table 3 brainsci-13-00205-t003:** Comparison of blood routine data between MSA and HC.

Indices	Participants		Effect Size	*p* Value
	MSA (*N* = 26)	HC (*N* = 27)		
WBC count (×10^9^/L)	5.59 (4.55, 6.37)	5.71 (4.53, 6.30)	−0.187	0.852 ^b^
Neutrophils (%)	55.80 (6.85)	56.92 (8.55)	−0.526	0.601 ^a^
Neutrophils (×10^9^/L)	3.03 (2.51, 3.65)	3.31 (2.38, 3.93)	−0.472	0.637 ^b^
Lymphocytes (%)	32.61 (5.94)	31.89 (7.69)	0.382	0.704 ^a^
Lymphocytes (×10^9^/L)	1.90 (1.41, 2.00)	1.71 (1.47, 2.11)	−0.044	0.965 ^b^
NLR	1.65 (1.40, 2.05)	1.82 (1.35, 2.23)	−0.427	0.669 ^b^
Monocytes (%)	7.35 (6.10, 8.55)	7.30 (6.80, 8.60)	−0.463	0.643 ^b^
Monocytes (×10^9^/L)	0.41 (0.34, 0.52)	0.44 (0.35, 0.51)	−0.596	0.551 ^b^
Eosinophils (%)	2.35 (1.43, 4.05)	2.70 (1.60, 4.10)	−0.525	0.599 ^b^
Eosinophils (×10^9^/L)	0.11 (0.08, 0.26)	0.14 (0.11, 0.24)	−0.758	0.449 ^b^
Basophils (%)	0.40 (0.20, 0.63)	0.40 (0.30, 0.60)	−0.333	0.739 ^b^
Basophils (×10^9^/L)	0.02 (0.01, 0.03)	0.02 (0.02, 0.04)	−0.431	0.667 ^b^

Data presented as mean (SD) or median (25% quantile, 75% quantile). *p* value calculated by Student’s *t*-test ^a^ or Mann–Whitney U test ^b^. Abbreviations: MSA: multiple system atrophy; HC: healthy controls; WBC: white blood cell; NLR: neutrophils-to-lymphocytes ratio; SD: standard deviation.

**Table 4 brainsci-13-00205-t004:** Comparison of lymphocyte subsets data between MSA and HC.

Indices	Participants		Effect Size	*p* Value
	MSA (*N* = 26)	HC (*N* = 56)		
Lymphocyte Subsets				
Total T cells (CD3^+^CD19^−^) (%)	70.75 (64.18, 75.48)	75.40 (70.19, 78.32)	−2.307	0.021 ^b^
Total T cell count (CD3^+^CD19^−^) (/μL)	1185.12 (281.31)	1238.93 (358.75)	−0.674	0.502 ^a^
Total B cells (CD3^−^CD19^+^) (%)	12.54 (9.28, 16.03)	12.05 (8.67, 15.07)	−0.623	0.533 ^b^
Total B cell count (CD3^−^CD19^+^) (/μL)	202.50 (136.50, 307.50)	183.00 (127.50, 297.75)	−0.598	0.550 ^b^
T/B (×10^9^/L)	5.85 (3.90, 7.49)	6.40 (4.58, 8.47)	−0.947	0.344 ^b^
NK cells (CD3^−^CD16^+^CD56^+^) (%)	16.11 (11.71, 20.95)	12.09 (7.56, 16.34)	−2.342	0.019 ^b^
NK cell count (CD3^−^CD16^+^CD56^+^) (/μL)	246.50 (180.00, 341.25)	192.50 (125.50, 266.75)	−2.397	0.017 ^b^
T+B+NK (%)	99.34 (99.03, 99.58)	99.47 (98.87, 99.69)	−0.444	0.657 ^b^
T+B+NK cell count (/μL)	1512.50 (1348.25, 1978.00)	1622.50 (1424.25, 1945.00)	−0.204	0.838 ^b^
T cells Subsets				
Th cells (CD3^+^CD4^+^) (%)	46.13 (6.00)	42.47 (8.95)	2.180	0.033 ^a^
Th cell count (CD3^+^CD4^+^) (/μL)	789.85 (231.21)	719.45 (251.19)	1.210	0.230 ^a^
Tc cells (CD3^+^CD8^+^) (%)	20.92 (6.39)	25.62 (8.00)	−2.629	0.010 ^a^
Tc cell count (CD3^+^CD8^+^) (/μL)	326.50 (272.75, 416.25)	391.00 (327.50, 531.50)	−2.143	0.032 ^b^
Th/Tc	2.16 (1.72, 3.22)	1.71 (1.17, 2.47)	−2.556	0.011 ^b^
Th (CD3^+^CD4^+^CD28^+^)/Th (%)	97.77 (93.79, 98.80)	94.69 (89.47, 97.31)	−2.362	0.018 ^b^
Tc (CD3^+^CD8^+^CD28^+^)/Tc (%)	63.29 (15.11)	54.71 (15.99)	2.299	0.024 ^a^
Activated T cells (CD3^+^HLA-DR^+^) (%)	14.90 (10.37, 16.47)	19.94 (14.98, 26.54)	−3.398	0.001 ^b^
Activated Tc cells (CD3^+^CD8^+^HLA-DR^+^)/Tc (%)	40.84 (24.40, 46.51)	37.92 (29.01, 48.53)	−0.538	0.591 ^b^
Treg (%) (CD3^+^CD4^+^CD25^+^CD127low^+^)	3.79 (1.26)	3.26 (1.33)	1.689	0.095 ^a^
Natural Treg (%) (CD45RA^+^CD3^+^CD4^+^CD25^+^CD127low^+^)	0.95 (0.44)	0.90 (0.45)	0.482	0.631 ^a^
Induced Treg (%) (CD45RO^+^CD3^+^CD4^+^CD25^+^CD127low^+^)	2.90 (1.88, 3.64)	2.22 (1.34, 3.38)	−1.859	0.063 ^b^
Induced Treg/Natural Treg	0.35 (0.17)	0.40 (0.18)	−1.090	0.279 ^a^

Data presented as mean (SD) or median (25% quantile, 75% quantile). *p* value calculated by Student’s *t*-test ^a^ or Mann–Whitney U test ^b^. Abbreviations: MSA: multiple system atrophy; HC: healthy controls; Th cell: helper T cell; Tc cell: cytotoxic T cell; NK cell: natural killer cell; Treg: regulatory T cell; SD: standard deviation.

**Table 5 brainsci-13-00205-t005:** Comparison of cytokines between MSA and HC.

Cytokines	Participants		Effect Size	*p* Value
	MSA (*N* = 21)	HC (*N* = 20)		
IL-1β (pg/mL)	5.00 (5.00, 6.50)	5.00 (5.00, 19.95)	−1.265	0.206
IL-2R (U/mL)	327.00 (246.50, 443.50)	341.00 (246.75, 408.75)	−0.261	0.794
IL-6 (pg/mL)	2.27 (1.50, 3.62)	1.51 (1.50, 3.46)	−1.076	0.282
IL-8 (pg/mL)	13.80 (9.05, 47.70)	15.90 (9.03, 37.00)	−0.339	0.735
IL-10 (pg/mL)	5.00 (5.00, 5.00)	5.00 (5.00, 5.00)	−0.976	0.329
TNF-α (pg/mL)	8.20 (6.50, 14.90)	9.25 (6.65, 16.65)	−0.417	0.676

Data presented as median (25% quantile, 75% quantile). *p* value calculated by Mann–Whitney U test. Abbreviations: MSA: multiple system atrophy; HC: healthy controls; IL: interleukin; TNF-α: tumor necrosis factor alpha; SD: standard deviation.

## Data Availability

The original datasets in this study are available from the corresponding author upon reasonable request.

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
