# Peer review of "The Peripheral Immune Traits Changed in Patients with Multiple System Atrophy"

_brainsci, 2023, doi:10.3390/brainsci13020205_

Round 1
Reviewer 1 Report
This manuscript covers an interesting research topic, The Peripheral Immune Traits Changed in Patients with Multiple System Atrophy. Overall, the manuscript has some strengths, but also various weaknesses, as outlined below. I have suggestions below as to how the manuscript could be improved.
1. The authors do not make an adequate case for why this article should be published, when other similar and more extensive article (higher sample size) has been recently published, including: Cao et al., 2020 (PMID: 32733370); the author should briefly discuss more about its relevancy and conclude it to more meaningful outcomes.
2. This study has low statistical power, as per Jewell (2004) Statistics for Epidemiology, and article published by Setia (2016) “Methodology Series Module 2: Case-control Studies” (PMID: 27057012), the optimal ratio is 4 controls: 1 case, to increase in statistical power.
3. Author should perform the effect size analysis.
4. Recent important literatures were not cited for example “Heterogeneity of Multiple System Atrophy: An Update, 2022” (PMID: 35327402), Evidence for Peripheral Immune Activation in Parkinson’s Disease (PMID: 33994989)
5. Serum immune markers were not well considered in manuscript, author should consider IFNγ as well as macrophage migration inhibitory factor (PMID: 21550814), for peripheral immune in MSA patients.
6. The study was conducted in both male and female, but results were not presented as sex stratified. It should be explained on gender basis as well.
Reviewer 2 Report
This is interesting manuscript in the area of ​​immune response involvement in multiple system atrophy.
However, as also as highlighted by the authors themselves, the study has some limitations (4.4. Limitations).
These limitations should be overcome, mainly by increasing the sample size, and by distinguishing the disease subtypes and better classify the patient groups by severity of symptoms.
In addition, further studies should be added, as follow described:
2.3. Serum cytokine detection
Activated NK cells secrete a wide variety of cytokines, such as IFN-γ, TNF-α, GM-CSF, IL-10, IL-5, and IL-13 and chemokines such as MIP-1α, MIP-1β, IL-8, and RANTES. In particular, IFN-γ has been shown to be one of the most potent effector cytokines secreted by NK cells )Paolini R, Cytokine Growth Factor Rev, 2015).
Consequently, in addition to cytokines already detected, a most complete cytokine release assay could give more detailed information.
2.4. Search strategy for literature review
It would be interesting to add the search term “Inflammasome” 4.1 Increased NK and alpha-syn patology
Take in consideration this article (and others similar): NK cells clear α-synuclein and the depletion of NK cells exacerbates synuclein pathology in a mouse model of α-synucleinopathy.
“Recently, a study demonstrated that natural killer cells scavenge alpha-synuclein aggregates, the primary component of Lewy bodies, and systemic depletion of natural killer cells results in exacerbated neuropathology in a mouse model of alpha-synucleinopathy, making them a highly relevant cell type in Parkinson's disease. However, the exact role of natural killer cells in Parkinson's disease remains elusive” (Rachael H. Earls, 2020).
At light of above described, since NK depletion in PD mouse models showed to exacerbate several symptoms, the percentages and absolute numbers of NK detected should be compared with the stages and the severity of the symptoms.
A monitoring over time of inflammatory markers and the changes of peripheral immune traits can be also useful to better understand the real involvement of NK cells and helper T cells in the pathology of MSA.
In conclusion, it is a very interesting research, but still incomplete and it deserves further analysis in order to allow the formulation of valid hypotheses.
Round 2
Reviewer 1 Report
The author should include a section where they need to mention about the weakness of studies, for example: low sample size, missing some important cytokines analysis etc.
Reviewer 2 Report
It would be more interesting too carry out future studies including different population groups and with a more complete assessment of inflammatory markers A comparative analysis with PD affected patients could be also useful. Author Response
Please see the attachment.
